# The Revolution in Breast Cancer Diagnostics: From Visual Inspection of Histopathology Slides to Using Desktop Tissue Analysers for Automated Nanomechanical Profiling of Tumours

**DOI:** 10.3390/bioengineering11030237

**Published:** 2024-02-28

**Authors:** Martin Stolz

**Affiliations:** National Centre for Advanced Tribology at Southampton, Faculty of Engineering and Physical Sciences, University of Southampton, Southampton SO17 1BJ, UK; m.stolz@soton.ac.uk

**Keywords:** cancer, biomarker, atomic force microscope, artificial intelligence, mechanobiology, MEMS sensor, IT-AFM

## Abstract

We aim to develop new portable desktop tissue analysers (DTAs) to provide fast, low-cost, and precise test results for fast nanomechanical profiling of tumours. This paper will explain the reasoning for choosing indentation-type atomic force microscopy (IT-AFM) to reveal the functional details of cancer. Determining the subtype, cancer stage, and prognosis will be possible, which aids in choosing the best treatment. DTAs are based on fast IT-AFM at the size of a small box that can be made for a low budget compared to other clinical imaging tools. The DTAs can work in remote areas and all parts of the world. There are a number of direct benefits: First, it is no longer needed to wait a week for the pathology report as the test will only take 10 min. Second, it avoids the complicated steps of making histopathology slides and saves costs of labour. Third, computers and robots are more consistent, more reliable, and more economical than human workers which may result in fewer diagnostic errors. Fourth, the IT-AFM analysis is capable of distinguishing between various cancer subtypes. Fifth, the IT-AFM analysis could reveal new insights about why immunotherapy fails. Sixth, IT-AFM may provide new insights into the neoadjuvant treatment response. Seventh, the healthcare system saves money by reducing diagnostic backlogs. Eighth, the results are stored on a central server and can be accessed to develop strategies to prevent cancer. To bring the IT-AFM technology from the bench to the operation theatre, a fast IT-AFM sensor needs to be developed and integrated into the DTAs.

## 1. The Need for Improved Breast Cancer Diagnosis

According to the World Health Organization, cancer is a leading cause of death globally, and in 2020 alone, an estimated 10 million people died from it. Breast cancer is the most common type of cancer in women worldwide, accounting for about 25% of all cancer cases and 15% of cancer deaths [1]. It is estimated that 1 in 8 women in the Western world will develop breast cancer in their lifetime, which corresponds to around 4700 women in the UK who are diagnosed with breast cancer every month [2,3]. Cancer progression is an ever-evolving process that comprises various changes in the shape, structure, and interactions of cancer cells and the microenvironment in which they become more aggressive and spread to other parts of the body. During cancer initiation, progression, and metastasis, the extracellular matrix (ECM) of the tumour undergoes structural and functional changes. Cancer progression provides clues about the nature, behaviour, and aggressiveness of cancer that can be accessed by clinico-pathological markers such as tumour size, grade, stage, lymph node status, and hormone receptor status, which are used to measure the severity of a patient’s cancer. Cancer diagnosis requires histopathological examination, imaging methods, and genetic analysis, but none of these methods alone is enough to fully examine cancer [4,5]

## 2. Histopathological Examination of Breast Cancer

The pathologist’s most important method to assess cancer and its aggressiveness is by examining histopathology slides under a microscope. Preparing a tissue sample for histopathological analysis and diagnosis involves multiple steps carried out by laboratory personnel. Firstly, the tissue sample undergoes fixation, which involves immersing it in a chemical solution such as formalin or glutaraldehyde, preserving the tissue and preventing decay and degradation. Secondly, the tissue sample is dehydrated by transferring it through a series of increasing concentrations of alcohol, which removes the water from the tissue and makes it compatible with the embedding medium. Next, the tissue sample is cut into thin slices, a few micrometres thick, using a microtome. These slices are then transferred to a microscopy slide. Lastly, the tissue sections are treated with a combination of haematoxylin and eosin (H&E) which stains the nuclei purple and the cytoplasm pink.

Histopathological analysis of tumour lesions has a fundamental limitation as the tissue sections are only 2D, dried, artificially stained representations of the 3D living tissue. Image data, such as histopathological sections, magnetic resonance imaging (MRI) scans, or positron emission tomography (PET) scans, can provide information about cellular and tissue structures, such as cell type, morphology, and density. Still, it fails to determine the functioning of the native tissue in health and disease. An analogy might clarify this point: A new bridge can look impressive (structure) but knowing if it is properly functioning can only be validated once tested if the bridge can carry traffic across an obstacle, such as a river, a valley, or a road. The function of a biological entity is more important than how it looks. Histopathological analysis and other image modalities only provide structural data and cannot assess cancer function and progression and, therefore, need to be complemented by other types of clinico-pathological markers.

Analysing tumour lesions through histopathological analysis—examining tissue samples and producing the pathology report—requires specialised equipment and materials. It is a labour-intensive and complex process that can take sometimes over a week to complete. In the process of histopathological analysis and diagnosis, artefacts can be introduced in the tissue’s appearance, which can impact the result’s accuracy and can lead to errors when characterising the disease stage. The error rate related to anatomic pathology of breast cancer diagnosis is about 3% [6]. To eliminate human error, tumour analysis could benefit from automation, as other fields have, such as car production.

In addition to histopathological analysis, the Breast Imaging Reporting and Data System (BI-RADS) is often used to provide reports for breast imaging diagnoses such as mammography, X-ray, ultrasound, PET scan, and MRI, further demonstrating the limitations of clinical imaging data.

## 3. Using the Standardised BI-RADS Scoring System

The BI-RADS is a standardised scoring system for mammography used for X-rays, MRIs, or any other imaging modality. It was introduced by the American College of Radiology in 1993 [7]. The BI-RADS scoring system ranges from 0 to 6. BI-RADS 0 means that the screening needs to be repeated because the mammogram was inconclusive. BI-RADS 1 means negative, suggesting no lesions of any kind; this is the most common result for mammograms with no risk of malignancy. BI-RADS 2 shows benign masses such as fibroadenoma or fatty tissue, which are not cancerous; it carries about a 0% relative risk of harbouring some malignancy in there. A score of BI-RADS 3 is probably benign but carries some risk for some sort of very small latency, about 1 to 2%; this means that the finding is likely harmless, but there is a slight chance that it could be a delayed or hidden cancer, which could be a prompt for the patient to have a follow-up in the next six months to examine if the score can be downgraded to a BI-RADS 2 or upgraded to a BI-RADS 4. BI-RADS 4 is suspicious but not exactly known. The subcategories are 4A, 4B, and 4C, and they have different chances of being cancerous. The risk is around 2% to 10% for 4A, 10% to 50% for 4B, and 50% to 95% for 4C. BI-RADS 5 means that the finding has a very high probability, 95%, of being cancer, and malignancy is almost certain. BI-RADS 6 indicates known cancer [7].

One reason for these uncertainties may be the low spatial resolution of imaging methods such as mammography and MRI, which do not have nanometre resolution. Radiography (X-ray), CT, MRI, and ultrasound imaging (US) are frequently used for clinical imaging. The spatial resolution of these methods is usually several tenths of a micrometre or less [8]. These methods can image lesions, but they do not distinguish minor differences at the molecular level. MRI aids in finding suspicious areas, but due to its limited spatial resolution, it is not used as a screening tool for breast cancer [9].

The clinical imaging tools have another limitation: they do not produce any mechanical data, and they are only capable of providing images. The exception is some modes in ultrasound, where the resolution is limited by diffraction, which is usually about half a wavelength. Clinical ultrasound applications use wavelengths between 200 µm and 1 mm; the resolution is typically in the range of 100 to 500 µm. To surpass the diffraction limit and reach resolutions of 10 µm, some super-resolution methods have been created [10], which are still low-resolution relative to atomic force microscopy (AFM). Therefore, it is important to develop technologies that are more precise allowing the determination of the cancer subtypes especially in the BI-RADS 4 categories.

Over the past two decades, a medical technology has been developed named indentation-type atomic force microscopy (IT-AFM) [11,12,13]. This innovative technology is based on the atomic force microscope, providing high-resolution quantitative information about the nanomechanical properties and behaviours of cells and collagen-rich tissues. IT-AFM uses a nanometre-sized sharp tip that palpates the tissue following an array pattern to sample force curves along the biopsy to monitor the nanomechanical responses of the tissue. The obtained nanomechanical signature is different for healthy tissue, benign lesions, and malignant lesions and provides an in-depth view of cancer at the subcellular level, for obtaining a comprehensive view of a tissue’s functioning in both health and disease. Nanomechanical profiling of fresh biopsies and visual inspection of 2D sections of dehydrated tissues are complementary methods that can provide different types of information for the diagnosis of diseases such as cancer. Nanomechanical profiling can offer more information about the mechanical and functional aspects of the tissue, while visual inspection can offer more information about the histopathological and morphological aspects of the tissue. Combining these methods will improve the accuracy and reliability of diagnosis, as well as the understanding of the mechanisms and dynamics of disease development and progression. In the ideal case, nanomechanical profiling provides comprehensive information about the disease state of the tissue.

The IT-AFM method is a revolutionary way to detect and treat tissue disorders, by examining how they look and work at the level of molecules, on fresh samples. The technology can produce nanoscale (molecular) resolution and mechanical (functional) data that enable the creation of time-resolved databases (movies) of cancer disease progression from the early to the late stages. This may facilitate an automated cancer diagnosis.

## 4. High-Resolution IT-AFM Has Great Potential for Deciphering Biological Mechanisms

The sharp tip in combination with AFM’s high-dimensional sensitivity can provide the ‘eyes’ and ‘fingers’ to image, measure, and manipulate biological tissues at the nanoscale, which promises diagnostic and treatment potential. IT-AFM employs tips at the nanometre scale either for imaging or as probes for recording the mechanical response while loading/unloading a biological tissue. For high-resolution imaging, a sharp nanometre-sized tip at the end of a long cantilever contours the sample surface, as the tip is scanned line by line over the specimen landscape. In this way, the AFM creates a three-dimensional (3D) image of the topography. Moreover, when placed directly atop a sample the imaging tip can measure the physical properties of the tissue under the compressive load supplied by that tip. When taking force curves, the tissue is pushed against the tip while an optical deflection sensor records the bending (deflection) of the cantilever. Figure 1 shows the tissue stiffness profiles that result from force-mapping the tissue.

The micrometre-sized tip measures the stiffness of the tissue as shown in Figure 1a, but it cannot distinguish the fine-structural elements of the tissue. Only the nanometre-sized AFM tip can distinguish the different elements of the extracellular matrix when nanomechanically profiling tissues, as shown in Figure 1b,c. IT-AFM reveals two separate peaks in cartilage, one for the proteoglycan moiety and one for the collagen meshwork [11,12], while the nanostiffness profile in the structurally more complex tumour structures reveals a more multifaceted nanostiffness distribution [14,15]. As the disease progresses, these profiles alter and provide information of the disease progression that can be subsequentially monitored.

When measured at the micrometre scale, the articular cartilage stiffness is about 100 times higher than at the nanometre scale. As Figure 2a,b illustrate, the different levels of tissue structure that are being examined account for these differences—i.e., at the tissue level, the cartilage structural elements work together and resist the micrometre-sized spherical tip. At the nanometre level, distinct structural elements of the cartilage structure resist the indenter. The nanometre-sized tip senses the interaction with either the proteoglycan moiety or the individual collagen fibres. As shown in Figure 2c, the structural elements of a tumour and the tip have a more complex interaction. Plodinec et al. studied biological mechanisms by employing IT-AFM to investigate the “inverse correlation between healthy and cancer nanostiffness profiles” [14]. These measurements demonstrate the potential of IT-AFM, but the technology needs to be improved to provide more precise and routine nanostiffness profiling.

Unfortunately, the current AFM technology limits our ability to better understand the biological mechanisms and how the distinct elements and their interactions affect the tissue stiffness. As shown in Figure 1b, the nanostiffness peak of the proteoglycan moiety (bluish-purple peak) is 22.3 kPa and of the collagen meshwork (orange peak) is only 384 kPa, which seems to be too low. This may be because the chosen spring constant value for the cantilevers, k = 0.06 N/m, which was good to test the proteoglycan moiety of cartilage precisely (which has diagnostic significance), made the measurements of the collagen meshwork less accurate, as shown in the position of the peak and by the big standard deviation in the nanostiffness values. The indentation nanostiffness corresponds to the expected value of a gel made of proteoglycans (aggrecan) of a few kPa [16], while the bending stiffness of individual collagen fibres is expected to be in the low GPa range [17]. The stiffness also depends on the crosslinking of the fine-structural elements. A better understanding of the mechanism and how the elements contribute to tissue stiffness requires an improved sensor that can more accurately measure the stiffness of the elements from soft proteoglycans (kPa) to the significant stiffer collagen fibres (GPa). Experimental data combined with modelling will help understand tissue function in health and disease. To provide more precise nanostiffness data, our fast IT-AFM sensor design is based on a symmetric quasi-concertina (QC) spring with high linearity (see below), which means its output is proportional to its input over a wider range compared to conventional cantilevers that will make nanostiffness measurements ranging from the soft proteoglycan moiety to the much stiffer collagen meshwork more precise to allow interpretation of biological mechanisms.

## 5. IT-AFM Will Allow High-Precision Nanomechanical Evaluation Tools to Be Developed

To reduce costs in the health care system and to find the best products, there must be objective evaluation tools that are able to measure the functional properties in follow-up inspections. The FDA states that “It would be appropriate to characterise various mechanical properties at discrete time points following maturation in a suitable animal model.” [18]. Based on this suggestion, several pharmaceutical companies have expressed strong interest in the Desktop Tissue Analysers for screening purposes and in the arthroscopic AFM [19] for testing drugs in non-human primates. Obviously, the pharmaceutical industry is now looking for methods to meet possible changes in regulatory requirements. There is no other method beside IT-AFM to perform mechanical tests on the small joints of mice [12]. Indeed, a major limitation that prevents a rational understanding of the pathogenesis of osteoarthritis and hampers the development of treatments is the lack of sensitive tools to measure how substances influence osteoarthritis in clinical trials. Osteoarthritis is a disease that advances slowly, which makes drug development a costly and time-consuming process that involves long clinical trials. The reason we have no drugs for osteoarthritis is that the method of choice for clinical trials, measuring the joint space narrowing on X-rays, is not a precise or sensitive indicator of the cartilage loss or damage in osteoarthritis and therefore requires many patients. Moreover, given the limited time of patent protection, a faster assessment of disease-modifying drugs could be highly profitable for the pharmaceutical industry, while benefiting patients. Nanomechanical profiling with IT-AFM integrated into Desktop Tissue Analysers, the arthroscopic AFM, or catheter-based AFMs could be useful tools for developing drugs for osteoarthritis, cancer, and atherosclerosis [20], as it can help to understand the effects of drugs on the structure and mechanics of tissues. IT-AFM can also help to identify biomarkers of disease progression and response to therapy.

## 6. Using Genetic Analysis of Tissue Samples to Identify Breast Cancer in the Clinic

Some of the most common genetic tests in the diagnosis of breast cancer are the following: First, predictive genetic tests look for inherited mutations in genes that increase the risk of breast cancer, such as BRCA1, BRCA2, and TP53 [21]. These tests identify people with a high chance of developing breast cancer in the future. Second, diagnostic genetic tests that analyse the DNA of the tumour cells to find mutations may affect the growth, behaviour, or response to treatment of cancer. These tests help to choose the most effective therapy for each patient. Third, hormone receptor tests measure the levels of oestrogen and progesterone receptors on the surface of the tumour cells to help determine if hormone therapy is a suitable option for the patient [22]. Targeted therapy attacks the proteins or genes that distinguish them from normal cells and can be either combined with other treatments or used as the main treatment. A relatively new treatment option is immunotherapies that work effectively in some patients, especially for triple-negative breast cancer [23]. Unfortunately, immunotherapy is not effective for everyone and like the other treatments can have severe side-effects [24]. Fourth, HER2/neu tests measure the levels of a protein called HER2/neu on the surface of the tumour cells. If HER2/neu is overexpressed, targeted therapies with drugs are used to block the action of HER2/neu [25].

## 7. Breast Cancer Therapies and Open Research Questions

To treat breast cancer, one of the ways is to surgically remove the tumour, apply chemotherapy to stop the proliferation of cancer cells, or apply radiotherapy to eliminate cancer cells or prevent their recurrence [26]. Hormone therapy reduces or stops the hormones that make some breast cancers grow bigger and is given as the main treatment for advanced breast cancer or after surgery [27].

Breast cancer is a complex and heterogeneous disease that poses many challenges for research and treatment. Some of these challenges are to determine cancer subtypes, to understand the aggressiveness of cancer [28], neoadjuvant treatment response [29], or why most breast cancer patients do not respond to immunotherapy [30]. Another area of research is to develop predictive biomarkers for personalized medicine in breast cancer [31,32]. IT-AFM could be a useful research tool to assist in solving some of these issues.

## 8. Development of Fast IT-AFM for Clinical Tissue Diagnosis

Due to the structural simplicity of the tissue and flatness of healthy native articular cartilage surfaces, the IT-AFM technology was developed on articular cartilage to address osteoarthritis before tackling the structurally more complex tumours [14]. A key finding is that the AFM tip size is essential in measuring the properties of articular cartilage in health and arthritis. IT-AFM was compared with the results from instruments that use larger probes to test the stiffness of cartilage [33,34,35,36,37]. These clinical indentation testing devices test the cartilage with large probes and measure the average stiffness over big volumes, which obscures the small changes in cartilage properties. Hence, it is not feasible to detect differences between healthy and unhealthy types of arthritic cartilage using these large-scale clinical indenters [37]. Besides the possibility of using smaller indenter sizes, a driving factor for developing IT-AFM was our understanding that, in contrast to these classical clinical indenters, the actuators of an AFM exhibit much higher z-dimensional sensitivity of ~0.01 nm versus the ~1 μm provided by the (mechanical) clinical indenters. The question was which factor had more impact on the diagnostic value: the higher z-resolution or the size of the indenters.

Using IT-AFM, the elastic moduli of articular cartilage at two different levels of tissue organisation were measured and “demonstrated that the dynamic elastic modulus, E*, using spherical indenter tips (radius = ~2.5 μm) and sharp pyramidal tips (radius = ~20 nm) […] revealed average modulus was ~1.3 MPa, in agreement with available millimetre-scale data, whereas with the sharp pyramidal tips, it was typically 100-fold lower” [13]. We also detected the early changes in ageing and osteoarthritic cartilage [12] and we measured the changes in the nanostiffness signature of articular cartilage induced by changing the ionic strength of the PBS bathing solution [11]. This enlightens the IT-AFM to provide better clinical information due to the much higher z-resolution as well as the higher spatial resolution provided by the sharper nanometre-sized AFM tips [11,13,38]. Other studies have confirmed this finding on articular cartilage [39,40], and on breast tumours [14] and liver tumours [15].

The IT-AFM measurements allow the conclusion that the disease develops and progresses in a bottom-up progression: “The pathological changes in osteoarthritis […] start at the molecular scale and spread to the higher levels of the architecture of articular cartilage to cause progressive and irreversible structural and functional damage” [12]. The fast IT-AFM Desktop Tissue Analysers would facilitate answering those questions. However, this bottom-up progression underlines the necessity of using nanometre-sized indenters. Tip size is also an important aspect for developing clinical tools, as disease diagnosis and biopsy examination would be easier with larger microfabricated tips [11]. The arthroscopic AFM was the first attempt at translating the IT-AFM technology from the bench to a clinical application [19]. In summary, the early changes in diseases that affect collagen-rich tissues, such as arthritis, solid cancers, and atherosclerosis [41], can only be measured at the nanometre scale.

AFM has become a popular tool to study how the mechanical properties of tissues affect cancer development in laboratory settings [42,43,44,45,46,47]. It has been shown that cancer cells are highly deformable, allowing them to enter the bloodstream and spread to other organs, while tissues with cells that do not have such nanomechanical characteristics will not be able to metastasise [48]. The AFM has opened the exciting possibility of diagnosing cancer, which is key to developing effective therapies to slow or halt disease progression. Several attempts have been made to provide diagnostic tools that allow for cancer detection [49,50,51,52,53,54,55]. One interesting tool that aims to detect cancer is the Optics11Life device [56,57]. However, its micrometre-sized tips may be too big to measure the important clinically relevant information. The device also lacks the speed to map enough force curves to provide statistically relevant indentation data. Moreover, it may be easy and handy to use the pre-calibrated probes of Optics11Life, but it may become costly if the tips become dirty and need to be changed often, which is inevitable when probing biological tissues. The AFM does not need to use pre-calibrated probes because the calibration of the cantilevers’ spring constant has been integrated into modern AFMs. In addition, using small discs of polydimethylsiloxane (PDMS) with known elastic moduli could serve as calibration standards to obtain reproducible data, similar to the usage of agarose gels with known elastic moduli as reference standards as we have produced and utilized [11]. Table 1 summarizes how the IT-AFM method has been used for nanomechanical profiling of tissues.

As demonstrated on healthy, diseased, osteoarthritic, and engineered cartilage, mechanical profiling at the nanometre scale provides valuable insights into the tissue condition. By using IT-AFM nanomechanical profiling of fresh tumour samples at the nanometre scale, functional diagnostics may help to establish the cancer stage and subtype or estimate the patient’s treatment outcome.

## 9. Developing Desktop Tissue Analysers (DTAs) to Generate Time-Resolved Databases (Movies) of Cancer Disease Progression for Automated Cancer Diagnosis

So far, the as-described applications of IT-AFM only provide snapshots of the disease, as documented in osteoarthritis [12,13] and breast cancer [14]. When taking the nanomechanical profile of a lesion at one point in time, it is comparable to taking a snapshot of a movie, which makes understanding the storyline of the movie difficult, as it can miss or distort the context, the development, and the meaning of the movie. However, if multiple biopsies are taken over time, the disease progression can be tracked and reconstructed into movies, revealing the high-resolved functional profiles as time-resolved series of nanostiffness profiles that tell the story of the cancer progression. By using nanoscale resolution and mechanical testing time-resolved databases, the superior data help to choose the best therapy. The IT-AFM technology will improve the prognosis and treatment outcomes of the patients, by more accurately determining the stage and the features of the tumour. The movies can be generated by monitoring a sequence of multiple snapshots; for example, on subsequent time points from samples from a single patient. Since it is practically not feasible to take subsequent biopsies from a single patient, the information can also be obtained by generating databases that reflect cancer progression in different patients. Machine learning technologies are needed to sort the force curves according to the same cancer subtypes. To fully understand the spectrum of cancer subtypes that occur in tumours, complete sets of stiffness maps are required, from healthy to highly advanced cancer tissue. Once the movie for each cancer subtype is known, it then allows the use of this reference database to determine at which point in time the cancer has already progressed in the patient.

The databases require profiling at the nanometre scale to function. For IT-AFM tests, the DTAs can download these reference databases. The DTA then automatically finds the cancer subtype and the state of cancer progression. Since IT-AFM technology is driven by artificial intelligence (AI), it will evolve at different levels until the technology is fully automated. As with all AI systems, the extent of automation will progressively evolve. In the beginning, tests on tumour biopsies will only provide a second opinion to the surgeon and the sample can be integrated into the standard-care workflow to provide further tests. Ultimately, the system will not only work autonomously but also provide databases that may help to develop cancer prevention. As with other AI-based systems, its aim is to replace traditional histopathological analysis with nanomechanical profiling of fresh biopsies to diagnose breast cancer. Self-driving cars can have unlimited incidences, but the breast has only a limited number of structures. It needs to be seen if IT-AFM time-resolved reference databases of fresh biopsies can completely replace other clinico-pathological markers or if other markers, like genetic analysis, are still required to supplement nanostiffness data.

## 10. Single-Cell Mechanical Testing May Not be Very Effective for Cancer Diagnosis

As cancer cells metastasize, they become much softer, which makes it easier for them to spread [48,66,67,68]. Cross et al. found that cancer cells are 70% softer than normal cells and proposed that this feature could help identify and diagnose cancer [48,69]. The method they proposed involves 12 h of cell culturing followed by taking force curves on a single cell as well as on different cells to obtain reliable statistics. Besides such technical issues, single-cell methods for cancer diagnosis may not reflect the complexity and heterogeneity of tumour tissues and be of limited value [66].

A prominent device that allows the elastic properties of single cells to be measured is the optical stretcher. The device uses two laser beams to trap and deform cells in suspension, without touching them. The optical stretcher can measure the elasticity of cells [70,71]. However, since cells act differently in their tumour microenvironment than when they are isolated, any mechanical-based approach to detect cancer in single cells may have limited applications.

## 11. The Development of High-Speed Atomic Force Microscopy (HS-AFM)

The slow line scan speed of around 2 Hz per line for conventional AFMs is the result of the low resonance frequency of cantilevers for standard AFM ranging from 10 to 70 kHz, which are too low for high-speed probing.

Short cantilevers with higher resonance frequencies are needed for faster scanning. The short cantilevers are typically a few microns in length, about 20 times smaller than conventional cantilevers, and have a resonance frequency of 1.2 MHz, compared to the conventional AFM cantilever of only 18 kHz (Table 2). The resonance frequency determines the probing speed of the AFM. All force–volume maps (nanostiffness maps) in our publications related to the development of IT-AFM have been monitored at 3 Hz of cyclic loading [11,12,13]. The image acquisition by standard AFM at 2 Hz line scan and 512 lines takes 4 min, while force –volume mapping over an array of 32 × 32 takes about 12 min. Higher indentation rates are prevented by the hydrodynamic drag of the V-shaped cantilevers.

In contrast, the ultra-small cantilevers with reduced overall cantilever mass result in higher resonance frequencies which result in shorter response times and higher temporal resolutions [73,75,76,77]. HS-AFM can image biological samples successively at rates between 10 and 100 fps, as demonstrated by the imaging of actin filaments and microtubules at 25 fps near video rate [78,79]. It is tempting to employ short cantilevers for recording force –distance curves at high speeds [72]. Unfortunately, it is not feasible to use short cantilevers for imaging or indentation testing of tumour samples. For HS-AFM imaging, which employs Tapping Mode imaging, the sample needs to be flat to avoid losing contact between the tip and sample during high-speed scanning. For force–volume probing, the cantilevers need to provide a sufficient out-of-plane range to provide sufficient z-deflection. Plodinec et al. reported indentation depths up to 3000 nm on breast tumours [14]. It is not feasible to generate indents of 3 μm with a cantilever that only has a length of 10 μm.

While the high-speed AFM imaging method may not be directly transferable to the development towards a fast IT-AFM profiling Desktop Tissue Analyser for nanomechanical profiling of tumour biopsies, some aspects of the technological developments may be also relevant for fast IT-AFM nanostiffness mapping.

## 12. Which Modes of Operation Are Suitable for Fast IT-AFM?

Conceptually, the AFM is a simple device that is composed of several components, including a scanner, a cantilever with a sharp tip, a light source, a position-sensitive photodetector (PSPD), and an electronic feedback controller that maintains a given set-point. In a typical AFM setup, the laser reflects from the back of the cantilever onto the PSPD. The flexible cantilever, usually made of silicon nitride with a sharp tip at the free end, is brought into proximity with the sample, with a distance of just a few Angstroms, where it interacts with the sample surface due to the Van der Waal’s forces. This interaction force, typically in the nano-Newton (nN) range, causes a deflection of the flexible cantilever. The PSPD monitors and measures the amount of deflection in proportion to the strength of the interactions. A piezoelectric actuator enables the positioning of the sample in the lateral direction and the cantilever tip in the out-of-plane direction with very high-precision motions provided by the very high x,y-dimensional sensitivity (0.1 nm) and very high z-dimensional sensitivity of 0.01 nm. The AFM controller, through an electronic feedback loop, allows the regulation of the tip-sample interaction. The output of this feedback loop can be used to obtain either topographical (image) or physical information (nanostiffness). The basic AFM modes are contact mode, non-contact, Tapping Mode, and force–spectroscopy (Force–Volume) mode.

## 13. Contact Mode

In contact mode (CM), the sample surface is constantly in contact with the tip of the cantilever. This creates repulsive forces that are measured. The cantilever has a sharp tip on its lower side that interacts with the sample surface. The changes in the sample’s topography are typically detected employing the optical beam deflection (OBT) method that senses the out-of-plane deflection in the micro-machined cantilever caused by variations in the interaction forces between the sharp tip and the sample [80].

## 14. Non-Contact Mode and Intermittent Contact Modes

Non-contact and intermittent contact modes are either not in contact or in slight contact with the sample surface and are unsuitable for measuring nanostiffness on the tumour samples because the tip and sample have very short interaction during the vibration cycle while the tip only affects the upper layers of the sample as summarized below.

Non-contact mode (NCM) AFM measures the attractive Van Der Waals forces between the tip and a sample surface while the cantilever oscillates at its resonance frequency [81]. During the scanning process, the cantilever with a sharp tip oscillates above the sample surface at a set scan speed. To maintain a constant distance between the tip and the sample, changes in the cantilever’s phase, amplitude, or frequency are tracked, which are induced by the attractive forces (in the pico-Newton range). These small interaction forces allow for the imaging of soft samples without damaging them, but since it operates at a larger distance compared to intermittent AFM modes its resolution and scan speed are lower. NCM requires ultra-flat samples.

Tapping Mode AFM which was developed by Hansma et. al. [82] images the sample topography by lightly tapping the surface with the oscillating AFM tip, which uses the repulsive Van Der Waals forces. Since the cantilever’s oscillation amplitude changes with the sample surface topography, the topography image is obtained by monitoring these changes. Tapping Mode AFM is typically used to image samples that do not adhere well to the surface like DNA that would be swept away in contact mode AFM. Tapping Mode AFM requires samples that are much flatter compared to the tumour biopsies.

Both Peak Force Tapping (PFT) and Tapping Mode are both intermittent contact modes where the AFM tip touches the sample briefly, but they have different methods to regulate and quantify the force of contact. Tapping Mode operates close to the first resonant peak of the cantilever, while Peak Force Tapping operates in a non-resonant mode. PFT directly measures the force–distance curve of the cantilever probe as it approaches and retracts from the surface [83]. Peak Force tapping cannot be used for the mechanical testing of tumour specimens because of the high roughness of the tumour specimens.

Contact Resonance Imaging (CRI) was developed by Yamanaka et al. Tapping Mode in 1998 [84] who used it to measure the elastic modulus of thin films. In CRI mode, the sample is oscillated at the resonance frequency while the cantilever tip is in contact with the sample. CRI is useful as it can provide information about the nano-mechanical properties from very small volumes, but it will not allow it to be used for the testing of tumour samples.

At the time when we were developing the first prototype of the arthroscopic AFM [19], we thought that Pulsed Force Mode (PFM) would be a good addition to our device [85,86]. PFM, a non-resonant, intermittent-contact mode for AFM, modulates the z-piezo of the AFM sinusoidally, making the AFM tip touch and leave the sample surface repeatedly. In PFM, each pixel in the image has a force–distance curve that shows how the AFM tip and the sample interact. The nanostiffness of the sample can be calculated from this data. Like with the other modes mentioned in this paragraph, it would be desirable to simultaneously obtain nanostiffness properties from the simple imaging of topography at a scanning speed comparable to that in contact mode. In contrast to Tapping Mode where the physical contact with the sample surface is only ~1% of the time, PFM has much longer time in contact with the sample. We assumed that the long interaction time with the sample surface may allow for quantitative nanostiffness mapping, but we were not able to reproduce our IT-AFM results on articular cartilage [13] using the PFM. While non-contact mode, Tapping Mode, or Peak Force Quantitative Nano Mechanics (QNM) [87], and similar oscillatory AFM modes such as HarmoniX [88], have attempted to overcome the time-consuming limitations of Force–Volume mapping mode, they may not allow for probing the nanomechanical properties of the rough tumour samples.

## 15. Force–Spectroscopy and Force–Volume Mode

The force–spectroscopy mode is used to measure the force between the AFM tip and the sample at a single point. In Force–Volume mode, the force curves are measured and collected at multiple points over the entire sample area in a three-dimensional array, or “volume”, of force data. This map of the sample’s properties also provides a low-resolved topography image over the tested area (quasi-height image). The AFM Force–Volume mode may be the only AFM mode that is suitable for nanostiffness measurements of tumour samples. Towards bringing IT-AFM to clinical and pharmaceutical applications, high-resolution force mapping needs to be developed to allow monitoring of force maps with more than 100,000 force curves within a few minutes.

## 16. Tip Length and Geometry

Several methods have been employed for the fabrication of AFM tips, such as focused ion beam (FIB) milling [89], electron beam-induced deposition (EBID) [90], field emission-induced growth (FEIG) [91,92,93], or dielectrophoresis of multi-walled carbon nanotubes (MWCNT) [94]. To provide high-quality force maps on tumour samples, the AFM tip needs to be significantly longer than the AFM tip of ~5 μm on the very rough tumour samples [14]. Since the conventional pyramidal-shaped NPS tips (Table 1) quickly widen with indentation depth, the optimal diameter of the tip needs to be tested, but rod-like AFM tips with a diameter of probably 50 nm may be sufficient for cancer profiling. Besides being long, the AFM tips should best provide a conical shape with a high aspect ratio to prevent convolution between the tip and sample.

## 17. Fabrication of a Fast IT-AFM Sensor for the DTA

The IT-AFM method was developed based on Bruker’s (former Veeco) Multimode AFM [11,12,13]. The Multimode AFM was first introduced in 1993 as the world’s first commercial atomic force microscope. All measurements on articular cartilage were performed using the Multimode AFM and same cantilever type DNP-S (D) (Table 1). These ~205 μm long cantilevers have two major disadvantages. The first disadvantage is the slow acquisition time of 12 min for an array of 32 × 32 = 1024 force curves and 20 min × 12 min = 240 min (4 h) for mapping a tumour biopsy (see Methods [14]). The second disadvantage of conventional cantilever probes is related to the small AFM tip of ~5 μm. While the native surfaces of articular cartilage exhibit a sufficiently flat surface, the tumour biopsies are 100 times rougher than the height of the AFM tip of ~5 μm. The consequence is that the bottom surface of the cantilever frequently touches the sample before the tip reaches the surface resulting in disturbed indentation curves.

The fabrication of a fast IT-AFM sensor for the Desktop Tissue Analysers is performed in a wafer-scale batch process using silicon-on-insulator material. It consists of photolithography and electron-beam lithography, metallization, and micromachining of the cantilever [95]. We decided on a symmetric quasi-concertina (QC) spring that consists of a series of parallel beams connected by diagonal beams. It has high out-of-plane compliance and low in-plane compliance, which means that it can bend easily in the vertical direction but not in the horizontal direction. The QC spring has a high linearity, which means its output is proportional to its input. We integrated a Wheatstone bridge to improve the cantilevers’ sensitivity to piezoresistive deflection to lower the noise from thermal drifts and electronic fluctuations as have been used in various studies [96,97,98,99]. We chose a self-actuating readout that can be made in large quantities. The sensor will have active piezo resistors integrated to translate the motion of the tip into changes in voltage and will be robust and failsafe.

The small size of the sensor has a much higher Eigenfrequency, allowing for measurements to be conducted at a faster rate of up to approximately 1000 Hz (instead of 3 Hz) [95]. This reduces the acquisition time from several hours to approximately 10 min. The sensor will have a high resolution of 5.6 nN and a large range of 1000 µm, which means it can measure very small and large forces and displacements with high accuracy. In our current design, we avoided the problem of the cantilever touching the sample surface by using longer tips than the ~5 μm tips (Table 1) and positioned the tip in the middle of the quasi-concertina micro-electro-mechanical systems (MEMS) force–displacement sensor [95].

Another way to overcome the limited scan speed of the single cantilever is to use arrays of cantilevers to increase the throughput of fast IT-AFM measurements. The Millipede technology developed by IBM utilizes arrays of 1024 AFM tips in parallel for fast data storage [100,101], and could be adapted for fast parallel probing or the creation of multifunctional probes with added capabilities, such as using them as a nano-scalpel. Additionally, functionalization of the AFM tips could open up possibilities for administering small amounts of drugs to specific areas [20].

## 18. Outlook

The IT-AFM test provided by the AI-powered Desktop Tissue Analysers is compatible with the standard workflow for breast cancer diagnosis and the experience for the patient of having a biopsy will not change at all. After the IT-AFM test is complete, the tumour samples can be used for downstream measurements. Examining the biopsy with the DTAs, first jointly with a human physician and, ultimately, in a fully automated way, has several benefits. Once the technology is implemented and the time-resolved reference databases (movies) are available, it means that:IT-AFM tests are available within 10 min;No pathology report is needed which saves the cost of labour;Unlike MRI, CT, and mammography, the AFM can perform nanomechanical functional profiling, which may result in more detailed information and better treatment outcomes;IT-AFM analysis may aid breast cancer subtyping;IT-AFM could enhance the understanding of cancer aggressiveness and treatment response;The same high-quality tests can be performed by the DTAs in all parts of the world;Nanostiffness data are digital and stored in databases;The databases can be used to optimise treatment modalities;Developing cancer prevention can also benefit from the databases.

By enabling diagnosis that is faster, cheaper, and more accessible, as well as facilitating data sharing and collaboration among health professionals, IT-AFM technology will help to digitalize hospitals. The quality of life and well-being of millions of people worldwide could be improved by IT-AFM technology, which could reduce the time and cost of tissue analysis and thereby increase the accessibility and affordability of health care. The IT-AFM platform technology may be adapted for the analysis of other collagen-rich tissues that can be affected by diseases, such as arthritis, atherosclerosis, and solid cancers. IT-AFM is a platform technology that is customisable and scalable and will allow the generation of tissue databases in research, hospital, and pharma settings. As the renowned physicist and Nobel laureate Richard Feynman said in his 1959 lecture at the California Institute of Technology: “There is plenty of room at the bottom”. He continued his talk by saying that “there is no rule of physics that speaks against the possibility of a machine that can do anything”. Likewise, nothing in physics prevents the concept of the “movies” or the feasibility of what I am suggesting.

## Figures and Tables

**Figure 1 bioengineering-11-00237-f001:**
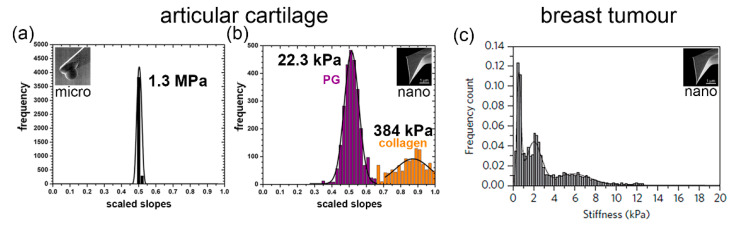
Nanomechanical profiles of articular cartilage and breast tumour. (**a**) The micrometer-sized tip cannot discriminate between the structural elements of the articular cartilage, resulting in a single peak and one value for the articular cartilage stiffness. (**b**) The proteoglycan moiety (bluish-purple) and collagen meshwork (orange) of the articular cartilage are distinguished by the nanometre-sized tip, which produces two peaks (bimodal distribution) and two corresponding nanostiffness values [11]. (**c**) Using a nanometre-sized tip, the nanomechanical profile of the tumour is more complex [14].

**Figure 2 bioengineering-11-00237-f002:**
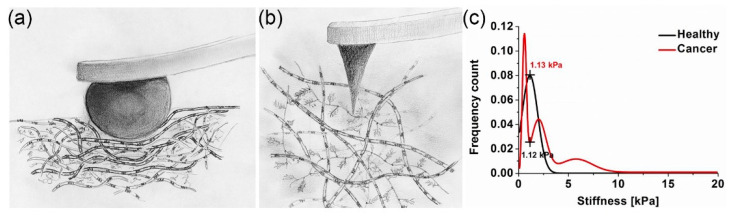
An illustration of the IT-AFM method and example of the biological mechanisms it studies. (**a**) Cartoon showing the interaction of a micrometre-sized tip (**a**) and a nanometre-sized tip (**b**) with cartilage [13]. (**c**) Inverse nanostiffness correlation between healthy and cancer biopsies [14].

**Table 1 bioengineering-11-00237-t001:** Table summary of IT-AFM work.

Authors	Tissue	Comment
Stolz et al. [58] (1999)	Porcine articular cartilage	First time showing the scale dependency of IT-AFM measurements
Stolz et al. [13] (2004)	Porcine articular cartilage	Demonstrates the scale dependency when assessing tissues
Stolz et al. [12] (2009)	Mouse articular cartilage Human articular cartilage	Osteoarthritis and ageing in mice deficient in type IX collagen, cartilage stiffness of osteoarthritic patients
Loparic et al. [11] (2010)	Porcine articular cartilage	Alterations of the nanostiffness profile of articular cartilage by varying the ionic strength of the PBS bathing solution
Grad et al. [59] (2012)	Engineered cartilage	Quality of engineered cartilage
Plodinec et al. [14] (2012)	Breast tumours	Nanomechanical signatures of breast cancer
Tian et al. [15] (2015)	Human liver	Nanomechanical signatures of liver cancer
Pang et al. [60] (2016)	Human articular cartilage	Effect of calcium ions on the nanostiffness of articular cartilage
Bouchonville et al. [61] (2019)	Soft and sticky materials	Technical paper on force curve analysis
Hartman et al. [39] (2020)	Human articular cartilage	Early detection of osteoarthritis
Muschter et al. [62] (2020)	Mouse articular cartilage	Mechanically induced osteoarthritis model
Zhang et al. [63] (2020)	Breast cancer	Modeling force curves
Fleischhauer et al. [64] (2022)	Mouse articular cartilage	Mechanically induced osteoarthritis
Rellmann [65] (2022)	ERp57 KO mice	Osteoarthritis

**Table 2 bioengineering-11-00237-t002:** Comparison between long and short cantilevers. Values for the long cantilevers (NPS-S, D) that were used for the papers related to the development of the IT-AFM method [11,12,13] and the short cantilevers (BL-AC10DS) that were used for the key papers by Ando et al. [72,73,74].

	Conventional AFM DNP-S (D)	HS-AFM BL-AC10DS
Cantilever type	Long cantilevers	Short cantilevers
Shape	V-shaped	Beam shaped
Length	205 μm	9–10 μm
Width	25 μm	2 μm
Thickness	5 μm	130 nm
Spring constant	k = 0.06 N/m	K = 0.1 N/m
Resonance frequency	f = 18 kHz	f = 1.2 MHz

## Data Availability

No new data were created or analyzed in this study. Data sharing is not applicable to this article.

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
