# Peer review of "The Revolution in Breast Cancer Diagnostics: From Visual Inspection of Histopathology Slides to Using Desktop Tissue Analysers for Automated Nanomechanical Profiling of Tumours"

_bioengineering, 2024, doi:10.3390/bioengineering11030237_

Round 1
Reviewer 1 Report
Comments and Suggestions for Authors
This paper provides an excellent overview about the use of IT AFM in fast cancer biopsy. Some minor editorials are
Line 33 : 1 in 8 women -> 1 of ?
Line 110 : Please write out the meaning of "US"
Line 144 - 149 : Suggestion to take it immediately after Line 133
Line 189 : write out the meaning of OA
Line 310 : is 3,000 nm 3 µm ? Komma irritates
Line 349 Reconsider wording from " I have summarized......." to "as summarized below"
Line 387 " I " and Line "388" harmonize to "we"
Line 420 Sentence starting with "First.." is confusing _ please re-write
Line 466 - 468 Please try to write these statements by marking them with bullets
Comments on the Quality of English LanguageEnglish is perfect
Reviewer 2 Report
Comments and Suggestions for Authors
The authors describe the development of portable desktop tissue analyzer based on indentation-type atomic force microscopy (IT-AFM) for fast nanomechanical profiling of tumors. The authors compare the current profiling of tumors such as H&E, MRI, PET, and X-Ray with IT-AFM to address its advantages. However, the authors do not provide figures that describe the principle of IT-AFM and profiling of tumors. Moreover, there are not a table summary the features that recent studies about IT-AFM.
Round 2
Reviewer 2 Report
Comments and Suggestions for Authors
The author should add a graph to describe the technology and biological mechanisms.
Author Response
Dear Reviewer,
> The author should add a graph to describe the technology and biological mechanisms.
I have included a new figure (Figure 2) and text to explain it (marked in red). I also needed to shift text to guarantee a logic flow.
Thank you!
Best wishes,
Martin
